# Spatiotemporal Heterogeneity and the Key Influencing Factors of PM_2.5_ and PM_10_ in Heilongjiang, China from 2014 to 2018

**DOI:** 10.3390/ijerph191811627

**Published:** 2022-09-15

**Authors:** Longhui Fu, Qibang Wang, Jianhui Li, Huiran Jin, Zhen Zhen, Qingbin Wei

**Affiliations:** 1School of Forestry, Northeast Forestry University, Harbin 150040, China; 2Key Laboratory of Forest Plant Ecology, Ministry of Education, Northeast Forestry University, Harbin 150040, China; 3School of Applied Engineering and Technology, Newark College of Engineering, New Jersey Institute of Technology, Newark, NJ 07102, USA; 4Department of Urban and Environmental Engineering, Ulsan National Institute of Science and Technology, Ulsan 44919, Korea; 5School of Geographical Sciences, Harbin Normal University, Harbin 150025, China

**Keywords:** PCA, GTWR, GWR, TWR, particulate matter, meteorological factors, NDVI

## Abstract

Particulate matter (PM) degrades air quality and negatively impacts human health. The spatial–temporal heterogeneity of PM (PM_2.5_ and PM_10_) concentration in Heilongjiang Province during 2014–2018 and the key impacting factors were investigated based on principal component analysis-based ordinary least square regression (PCA-OLS), PCA-based geographically weighted regression (PCA-GWR), PCA-based temporally weighted regression (PCA-TWR), and PCA-based geographically and temporally weighted regression (PCA-GTWR). Results showed that six principal components represented the temperature, wind speed, air pressure, atmospheric pollution, humidity, and vegetation cover factor, respectively, contributing 87% of original variables. All the local models (PCA-GWR, PCA-TWR, and PCA-GTWR) were superior to the global model (PCA-OLS), and PCA-GTWR has the best performance. PM had greater temporal than spatial heterogeneity due to seasonal periodicity. Air pollutants (i.e., SO_2_, NO_2_, and CO) and pressure were promoted whereas temperature, wind speed, and vegetation cover inhibited the PM concentration. The downward trend of annual PM concentration is obvious, especially after 2017, and the hot spot gradually changed from southwestern to southeastern cities. This study laid the foundation for precise local government prevention and control by addressing both excessive effect factors (i.e., meteorological factors, air pollutants, vegetation cover) and spatial-temporal heterogeneity of PM.

## 1. Introduction

Particulate matter (PM) is dispersed throughout the atmosphere, decreasing visibility, and interfering with plant photosynthesis [1,2,3,4]. In addition, numerous studies have demonstrated that airborne particulate matter has a negative impact on human health, increasing the chance of children’s breathing problems [5]. It has been established that PM_2.5_ and PM_10_ (atmospheric particles with aerodynamic diameters of 2.5 μm and 10 μm, respectively) are associated with an increased risk of human mortality [6]. According to the Global Burden of Disease 2010 comparative risk assessment (GBD), ambient air pollution from PM_2.5_ was ranked as the sixth highest overall risk factor for worldwide premature death [7]. 

According to the Global Urban Air Quality Index (AQI) Report, with the rapid urbanization and industrialization over the past three decades, Chinese cities have suffered serious air pollution challenges [8]. In January 2013, smog blanketed more than 1.3 million km^2^ of China, impacting approximately 850 million people [9]. In the same year, China released the Air Pollution Prevention and Control Action Plan, which established a quantifiable goal for decreasing PM_10_ concentrations in cities at the prefecture level and established a goal to reduce emissions by more than 10% by 2017, as compared to 2012 emission levels [10]. Some studies have claimed that China’s air pollution emissions have been reduced as a result of its reduction program, such as those examining Heilongjiang Province [4,11,12]. Except for Harbin and Suihua, the proportion of excellent days (the fraction of monitoring days with excellent and good ambient air quality indices in the effective monitoring days) in other cities of Heilongjiang Province exceed 90% in 2021 (http://sthj.hlj.gov.cn/kqzlxx/21080.jhtml (accessed on 12 September 2022)). Although the air quality in Heilongjiang Province has improved, further study is required to properly adopt more effective measures for the province. Furthermore, there have been minimal studies that have investigated air pollution in Heilongjiang Province compared to the economically developed regions in China, such as Beijing-Tianjin-Hebei, the Yangtze River Delta, and the Pearl River Delta [13,14,15,16,17]. In addition, the principal pollutants in Heilongjiang Province’s cities are inhalable particulate matter (PM_10_) and fine particulate matter (PM_2.5_) [18]. As a consequence, the particulate matter in Heilongjiang Province is the focus of this study (in this article, the particulate matter investigated are PM_2.5_ and PM_10_, which are referred to collectively as PM).

Both in China and abroad, numerous studies have conducted ecotoxicological analyses and chemical composition analyses of PM [19,20,21,22], while others have focused on the temporal and geographical characteristics of PM [23,24,25], as well as their influencing factors [26,27]. In general, both machine learning models and statistical models are commonly used to study the temporal and spatial distribution of PM. Artificial neural networks (ANNs), regression trees (RTs), support vector machines (SVMs), random forests (RFs), and XGBoost are classic examples of machine learning models. Although the accuracy of machine learning models has steadily increased in recent years, using a machine learning model to determine the link between explanatory and response variables is difficult [28,29,30,31]. In contrast, the statistical model can be used to investigate the relationship between explanatory and response variables. Global and local statistical models are the two types of statistical models. The global model applies a unique model for the entire study area; while the local model constructs various models for different times or locations and could explain the changes in the explanatory factors and the response variables through time and space. Ordinary least square regression (OLS) and geographically weighted regression (GWR) are one of the most classic global and local statistical models, respectively [32]. Typically, GWR uses a bandwidth and distance weight function to establish a model for a specific location. It can successfully cope with large-scale geographic data [33], and characterize the spatial non-stationarity of PM based on temporal stationarity. However, GWR may not be applicable to a large number of spatial occurrences over time. A recent study extended GWR to include a temporal component, and the extended GWR is otherwise known as the geographically and temporally weighted regression (GTWR) [34], which improves the fitting and forecasting accuracy by handling non-stationarity geographic data in time and space. When the space is stable, the spatial parameter of GTWR is neglected, so temporally weighted regression (TWR) was developed, which can successfully cope with geographic data throughout time [4]. Thus, the local model provides great potential for further investigating the relationship between PM and its influencing factors on both spatial and temporal scales. 

There are many factors that influence PM, including atmospheric pollutants (e.g., SO_2_, NO_2_, CO, etc.), meteorological factors (e.g., temperature, humidity, and wind speed), and vegetation factors [4,35,36,37]. Considering that Heilongjiang has a forest cover of 47.3% and a total forest area of 21.47 million hectares, the incorporation of vegetation factors would enhance model fitting. However, a large of variables would easily complicate the model and cause “the curse of dimensionality (or Hughes phenomenon [38])”, and high computational cost [39]. As a result, various methods are used to reduce the dimensions of the variables. A principal component analysis (PCA) is typically used to reduce the dimensions [40,41]. A PCA cannot only reduce the dimensionality but also retain more information. Its purpose is to construct a set of new orthogonal variables known as the principal components for replacing the original high-dimensional dataset [42,43,44,45]. 

This study integrates a principal component analysis (PCA) into the local models to efficiently investigate the temporal and spatial heterogeneity of PM (PM_2.5_ and PM_10_) and explains the key factors that influence it simultaneously. Specifically, the objectives include the following: (1) to conduct a PCA on the most relevant variables (including air pollutants, meteorological factors, and vegetation factors); (2) to establish a principal component analysis-based geographically weighted regression (PCA-GWR), a temporally weighted regression (PCA-TWR), and a geographically and temporally weighted regression (PCA-GTWR) and compare them to the corresponding global model, a principal component analysis-based ordinary least square regression (PCA-OLS); and (3) to explore the spatiotemporal characteristics of PM and the key influencing factors on them in Heilongjiang Province in recent years. This research provides a scientific foundation and technical support for a better understanding of the temporal and spatial heterogeneities of PM as well as the factors that drive PM, all of which are critical for future PM prevention and control.

## 2. Materials and Methods

### 2.1. Study Area

The present study was conducted in Heilongjiang Province, China, which is located in the northernmost region of China, within the longitudes of 121°11’ E to 135°05’ E and the latitudes of 43°26’ N to 53°33’ N (Figure 1). Heilongjiang has a mountainous topography in the northwest, north, and southeast and flat land in the northeast and southwest [4]. The climate of Heilongjiang Province is a cold temperate zone in continental monsoon climate. The major climatic features include warm and dry springs, hot and humid summers, dry autumns, and cold and lengthy winters, resulting in a protracted heating season. With a forest area of 21.47 million hectares and a forest coverage of 47.3%, Heilongjiang Province has a considerable forest area. The forest coverage rate of the different cities varies greatly, with some cities having greater forest coverage rates, such as Da Xing’an Mountain, Heihe, Yichun, and Mudanjiang, while others have lower forest coverage rates [46].

### 2.2. Data

#### 2.2.1. Air Pollutants Data

For this study, daily records of 6-criterion air pollutants from 2014 to 2018, including SO_2_, NO_2_, PM_10_, CO, O_3_, and PM_2.5_, were collected by the Environmental Monitoring Station of Heilongjiang Province based on the 57 environmental monitoring sites in Heilongjiang Province. The details of measuring air pollutants was described in Ambient air quality standards of P.R. China [47]. Apart from CO (mg/m^3^), all of the pollutants were measured in μg/m^3^.

#### 2.2.2. Meteorological Data

Sixteen meteorological variables were received from the Resource and Environmental Science and Data Center’s (http://www.resdc.cn/, (accessed on 12 September 2022)) daily data collection of meteorological station observations during the same time period. These included the average pressure (hPa), the maximum pressure (hPa), the minimum pressure (hPa), the average temperature (°C), the daily maximum temperature (°C), the daily minimum temperature (°C), the average relative humidity, the cumulative precipitation from 8 pm to the next-day 8 pm (daily cumulative precipitation, mm), the average wind speed (m/s), the maximum wind speed (m/s), the extreme wind speed (m/s), the sunshine hours (h), the average surface temperature (°C), the daily maximum surface temperature (°C), and the daily minimum surface temperature (°C). Maximum wind speed is the maximum of average 10-min wind speed in a given time period. Extreme wind speed is the maximum instantaneous wind speed in a given period of time.

#### 2.2.3. MODIS NDVI

The NASA Earth Observing System (EOS) Terra and Aqua satellites are equipped with the moderate resolution imaging spectroradiometer (MODIS) [48]. It includes two types of satellites (i.e., Terra and Aqua) and the Aqua satellite was used in this study.

The less cloudy MODIS L1B data of Heilongjiang Province with spatial resolutions of 1 km from 2014 to 2018 were selected and preprocessed (https://ladsweb.modaps.eosdis.nasa.gov/ (accessed on 12 September 2022)). First, radiometric calibration and a geometric correction were performed to correct and eliminate the “bow” effect and other deformations [49,50]. Then, the atmospheric correction of MODIS was performed and the atmospheric correction used the simplified dark pixel method [51]. Finally, the normalized difference vegetation index (NDVI), which captures the forest ecosystem’s structural information, was extracted using the binary classification approach. The NDVI equation is as follows [52]:(1)$$\mathrm{NDVI}=\frac{\left(IR-R\right)}{\left(IR+R\right)}$$
where *IR* and *R* is the pixel value in the infrared band and the red band, respectively.

The NDVI data were integrated with the air pollutants and meteorological data of 57 sites (i.e., 20 variables per site, one or two NDVI data per site per month, and 5945 records in total). The statistical indicators of the explanatory factors and the dependent variables are shown in Table 1. 

### 2.3. Methods

To reduce the dimensionality and multicollinearity of variables, a principal component analysis (PCA) was conducted on the most relevant variables (including air pollutants, meteorological factors, and vegetation factors). Based on PCA, the global model (PCA-OLS) and local models (PCA-GWR, PCA-GTWR, and PCA-TWR) were established. 

#### 2.3.1. Principal Component Analysis

Principal component analysis is a statistical analysis method that transforms multiple variables into a set of mutually orthogonal vectors (i.e., principal components). These principal components (PCs) retain most of the information of the original variables, which are usually expressed as the linear combination of the original variables. The relationship between principal component $y_{i}$ and original variable $x_{i}$ is as follows [42,53,54]:(2)$$y_{i}=a_{i1}x_{1}+a_{i2}x_{2}+a_{i3}x_{3}+\ldots +a_{ip}x_{p}\;,i=1,2\ldots ,p$$

Among them, $y_{i}$ is the *i*^th^ PC, $\left(x_{1},x_{2},x_{3},\ldots ,x_{p}\right)$ are the original variables, $a_{ij}$ represents the linear correlation coefficient for the *i*^th^ principal component and the *j*^th^ original variable, which is also known as the loading of the variable on the common factor. The PCs are independent of one another [55], so that:
(3)$$Cov\left(y_{i},y_{j}\right)=0\;\left(i\neq j,i,j=1,2,\ldots ,p\right)$$

Among them, $y_{i}$ and $y_{j}$ are the two PCs, and $Cov\left(y_{i},\;y_{j}\right)$ is the covariance between them. 

The variance of PCs decreases in turn, and the first PC contains the most information. the cumulative variance contribution rate is as follows:(4)$$c=\frac{\lambda_{k}}{\sum_{i=1}^{p}\lambda }\;\left(k=1,2,\ldots ,p\right)$$

Among them, $\lambda_{k}\left(k=1,2,\ldots ,p\right)$ is the variance of the $k^{th}$ PC, $\sum_{i=1}^{p}\lambda$ is the sum of variances for p variables.

The cumulative variance contribution rate of the first m variables is as follows: (5)$$b=\frac{\sum_{i=1}^{m}\lambda }{\sum_{i=1}^{p}\lambda }\;\left(m=1,2,\ldots ,p\right)$$

Among them, $\overset{m}{\underset{i=1}{\sum }}\lambda$ is the sum of variances for the first m variables. When b reaches a certain value (in this study, b is not less than 85%), the first m components are selected as PCs. To make the PCs interpretable, they are rotated using the maximum variance orthogonal rotation approach. 

Prior to PCA, the Kaiser-Meyer-Olikin (KMO) and Bartlett sphericity tests was used to determine whether the commonality of the variables was high. If KMO > 0.5 and p  < 0.01, the variables are suitable for PCA [54].

#### 2.3.2. Global Model (PCA-OLS)

The principal component analysis-based ordinary least square regression (PCA-OLS) is a global model used to describe the linear relationship between the response and independent variables [56], that is, PCs in this study. Its formula is as follows:(6)$$Y_{i}=\beta_{0}+\beta_{1}X_{1}+\beta_{2}X_{2}+\ldots +\beta_{k}X_{k}+\varepsilon_{i}\;\left(k=1,2,\ldots ,p-1,i=1,2,\ldots ,n\right)$$
where $Y_{i}$ is the response variable (i.e., PM_2.5_, PM_10_ in this study), $X_{k}\left(k=1,2,\ldots ,p-1\right)$ represents the effective or selected PCs. p is the total number of parameters to be estimated and n is the number of samples (in this study, *n* = 5945). $\beta_{k}$ is the model parameter, $\varepsilon_{i}\;$is the model error term, with an expected value of zero and a normal distribution. Since PCA-OLS is a global model, model parameters $\beta_{k}$ are estimated by using data of the entire study area and the model coefficient vector $\;\beta^{T}=\left[\beta_{0},\beta_{1},\beta_{2},\ldots ,\beta_{k}\right]\;$of PCA-OLS is estimated by
(7)$$\hat{\beta }={\left(X^{T}X\right)}^{-1}X^{T}Y$$
where X and Y are the vectors of PCs and PM, the superscript *T* denotes the transpose of a matrix.

#### 2.3.3. Local Models

##### PCA-GWR Model

To explore geographical nonstationarity, principal component analysis-based geographically weighted regression (PCA-GWR) is conducted in this study. PCA-GWR extends the PCA-OLS regression framework as follows [57]:(8)$$Y_{i}=\beta_{0}\left(u_{i},v_{i}\right)+\sum_{k=1}^{p-1}\beta_{k}\left(u_{i},v_{i}\right)X_{ik}+\varepsilon_{i}\;\;i=1,2,\ldots ,n$$
where $Y_{i}$ is the response variable (i.e., PM_2.5_, PM_10_ in this study), $\left(u_{i},v_{i}\right)$ is the spatial coordinates of position i. $\beta_{0}\left(u_{i},v_{i}\right)$ and $\beta_{k}\left(u_{i},v_{i}\right)\left(k=1,2,\ldots p-1\right)$ are the intercept and a set of *p* − 1 slope parameters at the i^th^ observation. $X_{ik}(k=1,2,$...,p−1) are a set of *p* − 1 PC at the i^th^ location, p is the total number of parameters to be estimated, $\;\varepsilon_{i}$ is the error term of i^th^ observation. 

$\beta \left(u_{i},v_{i}\right)$ could be estimated by the local least square method as follows [58]:(9)$$\hat{\beta }\left(u_{i},v_{i}\right)={\left[X^{T}W\left(u_{i},v_{i}\right)X\right]}^{-1}X^{T}W\left(u_{i},v_{i}\right)Y$$
where $\hat{\beta }\left(u_{i},v_{i}\right)$ is the vector of estimated coefficients for observation i, $W\left(u_{i},v_{i}\right)$ is a diagonal matrix in which the off-diagonal components are zero and the diagonal elements represent the geographic weights at the observation i, and the matrix is as follows:(10)$$W_{\left(u_{i},v_{i}\right)}=\left(\begin{array}{cccc} w_{i1} & 0 & \ldots & 0\\ 0 & w_{i2} & \ldots & 0\\ \ldots & \ldots & \ddots & \ldots \\ 0 & 0 & \ldots & w_{in} \end{array}\right)$$

Among them, $w_{ij}$ is spatial weight and it is determined by the spatial kernel function, also called a distance-decay function. 

A spatial kernel function is often classified into two types: fixed kernels and adaptive kernels [34]. The adaptive bisquare function is utilized to produce geographic weights in this study and the formula is as follows [59]: (11)$$w_{ij}=\left\{\begin{array}{c} {\left[1-{\left(\frac{d_{ij}}{h_{i}}\right)}^{2}\right]}^{2},\;if\;d_{ij}<h_{i}\\ 0,\;\;\;\;\;\;\;otherwise \end{array}\right.$$
where *h_i_* is a non-negative metric known as bandwidth, $d_{ij}$ is the distance between locations *i* and *j* and it is calculated as follows: (12)$${\left(d_{ij}^{S}\right)}^{2}=\lambda \left[{\left(u_{i}-u_{j}\right)}^{2}+{\left(v_{i}-v_{j}\right)}^{2}\right]+\mu {\left(t_{i}-t_{j}\right)}^{2}$$

As the distance between the two locations increases, the spatial effect between the two gradually decays and disappear beyond the bandwidth, that is, $w_{ij}=0$ [58]. The Cross-validation (CV) score and Akaike Information Criterion (AICc) criterion are standard metrics for determining optimal bandwidth [60,61].

##### PCA-GTWR and PCA-TWR Model

While temporal variability is considered, principal component analysis-based geographically and temporally weighted regression (PCA-GTWR) is conducted with a weight matrix among  i  and other points is calculated in spatial and temporal space. The PCA-GTWR model can be expressed as follows [34]:(13)$$Y_{i}=\beta_{0}\left(u_{i},v_{i},t_{i}\right)+\sum_{k=1}^{p-1}\beta_{k}\left(u_{i},v_{i},t_{i}\right)X_{ik}+\varepsilon_{i}\;i=1,2,\ldots ,n$$
where $Y_{i}$ is the response variable (i.e., PM_2.5_, PM_10_ in this study), $\left(u_{i},v_{i},t_{i}\right)$ are the spatial coordinates of position  i at specific time *t*. $\beta_{0}\left(u_{i},v_{i},t_{i}\right)$ and $\beta_{k}\left(u_{i},v_{i},t_{i}\right)\left(k=1,2,\ldots ,\;p-1\right)$ are the intercept and the parameter at the *i*^th^ observation with the spatio-temporal coordinate of $\left(u_{i},v_{i},t_{i}\right)$, respectively. $X_{ik}(k=1,2,$...,p) is the *k*^th^ PC at the i^th^ observation, $\varepsilon_{i}$ is the error term of observation *i*. The estimation of $\beta \left(u_{i},v_{i},t_{i}\right)$ is as follows:(14)$$\hat{\beta }\left(u_{i},v_{i},t_{i}\right)={\left[X^{T}W\left(u_{i},v_{i},t_{i}\right)X\right]}^{-1}X^{T}W\left(u_{i},v_{i},t_{i}\right)Y$$

The estimation of $\beta \left(u_{i},v_{i},t_{i}\right)$ is similar to that in PCA-GWR (Equation(8)). 

Because the spatio-temporal dimension is taken into account, a spatial–temporal weight matrix is constructed based on a spatio-temporal distance. In this study, spatio–temporal distance is as follows:(15)$${\left(d_{ij}^{ST}\right)}^{2}=\lambda \left[{\left(u_{i}-u_{j}\right)}^{2}+{\left(v_{i}-v_{j}\right)}^{2}\right]+\mu {\left(t_{i}-t_{j}\right)}^{2}$$
where  λ and μ are scaling factors used to balance the different effects of space and time, $d_{ij}^{ST}\;$ is the distance between point $\left(u_{i},v_{i},t_{i}\right)$ and point $\left(u_{j},v_{j},t_{j}\right)$.

In this study, an adaptive Gaussian distance–decay function was applied to construct the spatial–temporal weight matrix, and the formula is as follows:(16)$$w_{ij}=\exp\left\{-\frac{{\left(d_{ij}^{ST}\right)}^{2}}{h_{ST}^{2}}\right\}=\exp\left\{-\frac{{\left(d_{ij}^{S}\right)}^{2}}{h_{S}^{2}}\right\}\times \exp\left\{-\frac{{\left(d_{ij}^{T}\right)}^{2}}{h_{T}^{2}}\right\}$$
where $h_{ST}^{2}$ is a parameter of spatio-temporal bandwidth, and $h_{S}^{2}\;$ and $h_{T}^{2}$ are spatial and temporal bandwidth. When geographical variation is neglected (λ=0), Equation (15) is changed as follows:(17)$${\left(d_{ij}^{T}\right)}^{2}=\mu {\left(t_{i}-t_{j}\right)}^{2}$$
where $d_{ij}^{T}$ is the temporal distance. Based on $d_{ij}^{T}$, principal component analysis-based temporally weighted regression (PCA-TWR) is conducted [4]. PCA-TWR and PCA-GWR are special cases of PCA-GTWR without considering either spatial or temporal variation.

#### 2.3.4. Model Assessment

Akaike’s information criterion (AICc), adjusted coefficient of determination ($R_{a}^{2}$), root mean squared errors (RMSE), and mean absolute error (MAE) are used to assess the model performance. The formula of AICc is shown as follows:(18)$$\mathrm{AICc}=2n\ln(\hat{\sigma })+n\ln(2\pi )+n\left[\frac{n+tr\left(S\right)}{n-2-tr\left(S\right)}\right]$$

Among them, n is the number of samples, $\hat{\sigma }$ is the estimated standard deviation of the error term, and $\;tr\left(S\right)$ is the trace of hat matrix S(i.e., $\hat{Y}=SY$). In the GWR model, the hat matrix is as follows:(19)$$S=X{\left(X^{T}W\left(u_{i},v_{i}\right)X\right)}^{-1}X^{T}W\left(u_{i},v_{i}\right)$$

$R_{a}^{2}$ will not exaggerate the explained percentage compared to the coefficient of determination ($R^{2}$), and the formula is as follows:(20)$$R_{a}^{2}=1-\frac{\left(n-1\right)\left(1-R^{2}\right)}{\left(n-p\right)}$$
where n is the number of the sample, and p is the number of coefficients (including all predictor coefficients and the intercept).

The root mean squared errors (RMSE) and mean absolute errors (MAE) are also used in this study, and the formula is as follows: (21)$$\mathrm{RMSE}=\sqrt{\frac{\sum_{i=1}^{n}{\left(y_{i}-{\hat{y}}_{i}\right)}^{2}}{n}}$$
(22)$$\mathrm{MAE}=\frac{\sum_{i=1}^{n}\left|y_{i}-{\hat{y}}_{i}\right|}{n}$$
where $y_{i}$ is the observed values of the response variable and ${\hat{y}}_{i}$ is the predicted values of the response variable.

## 3. Results

### 3.1. Principal Component Analysis

The Kaiser-Meyer-Olikin (KMO) and Bartlett sphericity tests were conducted prior to the PCA, and the results (KMO = 0.76 > 0.5 and *p* < 0.01) revealed that the original 20 variables were highly correlated and PCA was appropriate. As shown in Table 2, the first six PCs were chosen due to the cumulative variance contribution rate of 87%. According to the principal component loadings, PC1~PC6 could represent the temperature, wind speed, air pressure, atmospheric pollution (SO_2_, NO_2_, CO), humidity, and vegetation cover from the original dataset, respectively.

### 3.2. Globel Model (PCA-OLS)

The stepwise PCA-OLS model for PM, a global model, was applied as a baseline for model comparison in this study. According to Table 3, the PM was negatively correlated with temperature (PC1), wind speed (PC2), and vegetation (PC6). PC5 was excluded for PM_10_ due to its nonsignificance. PM_2.5_ was positively correlated with pressure (PC3), atmospheric pollutants (PC4), and humidity (PC5), whereas PM_10_ was only positively correlated with pressure (PC3) and pollutants (PC4). The PCA-OLS models modestly fitted the data according to the $R_{a}^{2}$, i.e., 61% and 54% of the total variation of PM_2.5_ and PM_10_ can be explained by the PCA-OLS models. 

### 3.3. Local Models

To characterize the spatial and temporal heterogeneities, local models (i.e., PCA-GWR, PCA-TWR, and PCA-GTWR model) were utilized. The adaptive bisquare function was used, and AICc was employed to select a number of neighbors. Table 4 and Table 5 summarize the statistics of the parameter estimates of the PCA-GWR, PCA-TWR, and PCA-GTWR for PM_2.5_ and PM_10_, respectively. 

The sign of the global model parameters matches the average value of the local model parameters for both PM_2.5_ and PM_10_. It indicates that the local models follow the same trend as the corresponding global model: PM positively correlated with air pressure, other atmospheric pollutants, and humidity, while negatively correlated with temperature, wind, and vegetation cover. The number of neighbors of PCA-GWR is roughly twice that of the PCA-GTWR model, whereas that between PCA-GWR and PCA-GTWR is quite similar (i.e., around 600) for both PM_2.5_ and PM_10_. After considering spatial and temporal variation, PCA-GTWR performed better than PCA-GWR and PCA-TWR for PM according to the goodness-of-fit (i.e., $R_{a}^{2}$ and AICc). However, the local model of PM_2.5_ is superior (i.e., higher $R_{a}^{2}$) than the corresponding model of PM_10_.

### 3.4. Model Assessment

Table 6 depicts the goodness-of-fit of the PCA-based local and global models. No matter if it was the PM_2.5_ or PM_10_, PCA-GTWR performed the best (largest $R_{a}^{2}$, smallest AICc, RMSE and MAE), followed by PCA-GWR, PCA-TWR, and PCA-OLS. The PCA-TWR is much superior to the PCA-GWR, with a greater $R_{a}^{2}$, a lower AICc, RMSE, and MAE. This demonstrates that the PM’s temporal nonstationarity is more noticeable than its spatial nonstationarity. 

### 3.5. Spatial Characteristics of PCA-GTWR Parameter Estimates for PM

According to Figure 2, each PC factor had a difference influence in various locations for PM_2.5_. Temperature (Figure 2a) was the inhibitor of PM_2.5_ in most of Heilongjiang province. The inhibition was strong and obvious in the eastern area but weak in the west (e.g., Da Xing’an Mountain, Heihe, Qiqihar, Daqing, and Harbin). Wind speed (Figure 2b) primarily suppressed PM_2.5_, and the total inhibitory impact was comparable to temperature. Air pressure (Figure 2c) mostly promoted PM_2.5_, with a relative weak influence in Daqing, Harbin, Hegang, Jiamusi, and Shuangyashan. Atmospheric pollutants (Figure 2d) positively correlated with PM_2.5_, with three strata of correlations. The largest impact was observed the west of Harbin and the southern part around Daqing and Shuihua, whereas the smallest impact was observed in Da Xing’an Mountain, Heihe, and Qiqiha’er, and other places falling somewhere in the middle. Humidity (Figure 2e) had the ability to both enhance and inhibit PM_2.5_ levels. In the western cities, especially Harbin, Daqing, Suihua, and Qiqiha’er, humidity plays a role of promoting PM_2.5_; whereas in the eastern cities (e.g., Yichun, Hegang, Jiamusi, Shuangyashan, Qitaihe, and Jixi), it had a substantial negative effect. Vegetation cover (Figure 2f) obviously restricted PM_2.5_ concentration. The restriction trend decreased towards the east, with the smallest impact in Hegang, Jiamusi, and Shuangyashan.

Similar to PM_2.5_, different PC factors had various influences on PM_10_ in space (Figure 3). However, the magnitude of parameter estimates for PM_10_ was much larger than that for PM_2.5_. Temperature (Figure 3a) could both enhance and inhibit PM_10_. Temperature tended to promote PM_10_ in Da Xing’an Mountain, Harbin, and Heihe. Whereas in other places, it played an inhibitory role. The inhibitory impact was stronger in the eastern area (i.e., Hegang, Jiamusi, Shuangyashan, Jixi) than the western area (e.g., Qiqiha’er, Daqing, Suihua, and Yichun). Wind speed (Figure 3b) had an inhibitory influence on PM_10_, and the inhibitory effect was fairly uniform across the region. Both pressure (Figure 3c) and atmospheric pollutants (Figure 3d) positively correlated with PM_10_. The positive correlation between PM_10_ and pressure tended to decrease from south to north, while its correlation with atmospheric pollutants presented the opposite trend. Vegetation cover (Figure 3e) inhibited PM_10_, with strong suppression in the cities of Qiqihar, Daqing, Suihua, Harbin, and Mudanjiang, and the suppression became weaker towards the east.

### 3.6. Temporal Characteristics of PCA-GTWR Parameter Estimates for PM

Figure 4 shows that influence of each PC on PM has temporal non-stationarity and periodicity, particularly PC3 (air pressure) and PC4 (atmospheric pollutants). For PM, the GTWR parameter estimates of PC3 and PC4 in the heating (i.e., winter) and non-heating (i.e., spring, summer, autumn seasons) periods differed and periodically changed during the course of a year: high in the heating season (usually from October to April next year) and low in the non-heating seasons. This is because the higher concentrations of atmospheric pressure and other air pollutants in winter are correlated with higher PM concentrations than those in summer. 

The parameter estimates of PC1 (temperature) and PC2 (wind speed) has larger temporal periodicity for PM_10_ than PM_2.5_. The inhibitory effect of PC2 (wind speed) in autumn (September, October, and November) tends to be larger than that in other seasons. Due to the climate characteristics of Heilongjiang Province (cold temperate, and a temperate continental monsoon climate), the high wind speed in autumn tends to increase the inhibitory effect on PM. The inhibitory effect PC6 (vegetation coverage) on PM was obvious in 2014 but decrease and became stable afterwards. The impact of PC5 (humidity) on PM_2.5_ is not stable because PC5 is comprehensive factor that mainly controlled by daily average relative humility, daily cumulative humility and daily sun hours.

## 4. Discussion

### 4.1. The Key Influencing Factors of PM

Recently, many studies have investigated the connections between PM and a number of variables, including air pollutants, meteorological variables (such as temperature, wind speed, and relative humidity), the normalized difference vegetation index (NDVI) derived from satellite imagery, aerosol optical depth (AOD), human activities (such as transportation emission variables, the density of industrial plants, land use, GDP), and topographical factors and more [62,63,64,65,66,67]. Although it tends to improve PM predictions using more factors, it is difficult to remove multicollinearity among these factors and interpret the spatio-temporal relationship between PM and a number of factors. In this study, PCA was used to reduce the dimension of independent variables (i.e., from 20 original variables to 6 PCs) and remove the multicollinearity, and then OLS, GWR, TWR, and GTWR models were conducted based on the PCs to predict and explore the temporal and spatial heterogeneity of PM in Heilongjiang Province. Air pollutants (mainly SO_2_, NO_2_, and CO) represented by PC4 in this study have the largest influence on both PM_2.5_ and PM_10_ (see Table 2 and Table 3). Temperature expressed by PC1 had the second greatest impact on PM_2.5_, followed by air pressure (PC3), wind speed (PC2), vegetation cover (PC6), and humidity factor (PC5). Whereas the impacts of temperature, pressure, and vegetation cover on PM_10_ are roughly equivalent, followed by wind speed. 

Although PM_2.5_ and PM_10_ are closely correlated [68], this study still shows some differences in the correlation between PM_2.5_/PM_10_ and the influenced factors. The main distinction is that, in contrast to PM_2.5_, the humidity factor (PC5) is not significant for PM_10_ and is eliminated in the global model. This may be because the humidity factor in this study is a combined variable of daily average relative humidity (positive), cumulative precipitation (positive), and sun hours (negative). It is challenging to demonstrate a meaningful relationship between the mixed variable and the comparatively large particle (PM_10_). The second factor contributing to the difference was wind speed. The dilution effect of wind speed on PM_2.5_ is stronger than that of PM_10_ due to the relatively light quality, which is consistent with the previous study [68].

In general, PM is positively related to other air pollutants (mainly SO_2_, NO_2_, and CO). It is similar to the finding that 5 criteria air pollutants (i.e., PM_10_, SO_2_, NO_2_, CO, O_3_) had a positive impact on PM_2.5_, except O_3_, in the previous study [4]. The concentration of PM reduces as the temperature (PC1) rises. When the temperature is high, the atmospheric tropospheric motion becomes more intense, resulting in the upward transport of PM, and the high temperature encourages Brownian particle motion, which is more favorable to diffusion [26]. Not surprisingly, the wind speed (PC2) has a negative effect on PM, which is consistent with previous studies [69,70]. This is because that the higher the wind speed, the stronger the particle dispersion capacity and the lower the particle concentration [71]. Atmospheric pressure (PC3) plays a positive role in increasing PM centration. When an area is subjected to high pressure, air convection is reduced, allowing contaminants to accumulate more easily, and vice versa [26,72]. The ability of the forest’s complex canopy structure to absorb particulate matter has long been recognized as a critical tool for controlling PM [73,74]. In this study, vegetation cover (PC6) also has a considerable inhibitory influence on PM. The absolute magnitude of the coefficient indicates that it has a significant impact on preventing PM increases. The explanation could be that vegetation intercepts and absorbs particulate matter via Brownian diffusion, interception, and gravity deposition [75,76].

### 4.2. Global and Local Models

In the constructed model, the problem of the PCA-OLS model is that it uses a global model to represent the relationship between dependent and independent variables, ignoring the spatial or temporal effect of the studied variables. This spatial or temporal effect (spatial/temporal autocorrelation and heterogeneity) may violate the assumption of independent observation or constant variance, resulting in the biases of estimates of standard errors and imprecision of coefficient estimates [77,78]. PCA-GWR was better than PCA-OLS by considering spatial heterogeneity (11% and 17% improvement of adjusted $R^{2}$ of PM_2.5_ and PM_10_, respectively). PCA-TWR was also superior to PCA-OLS by considering temporal heterogeneity (23% and 28% improvement of adjusted $R^{2}$ of PM_2.5_ and PM_10_, respectively). The temporal heterogeneity is more significant than the spatial heterogeneity due to the obvious seasonal variation of PM, which is consistent with the previous study [4]. The PCA-GTWR model that considers both temporal and spatial heterogeneity has the best performance (compared with PCA-OLS, the adjusted $\;R^{2}$ increases by 25% and 31% for of PM_2.5_ and PM_10_, respectively). However, the improvements of PCA-GTWR from PCA-TWR are very limited, especially for PM_2.5_. It indicates that temporal information is more effective than spatial information for modeling PM based on PCA.

### 4.3. The Spatial and Temporal Distribution of PM Concentrations in Heilongjiang Province

Heilongjiang Province is located in northern China and is noted for its high latitude and harsh winters. Heilongjiang Province is one of China’s largest provinces, with a north-south latitude range of approximately 10° and an east-west longitude range of approximately 4°. According to Figure 5 and Figure 6, the PM content in the southern area is significantly higher whereas that in the northern region is rather low due to the large population and coal burning in the south. Furthermore, the PM concentrations remained high from 2014 to 2017, especially in 2017. From spatio-temporal perspective, the hot spot of PM_2.5_ concentration gradually changed from southwestern (e.g., Harbin, Daqing) to southeastern cities (e.g., Harbin, Mudanjiang, Qitaihe, and Jixi). According to Figure 5f and Figure 6f, it can be seen that the temporal distribution of PM in Heilongjiang Province has heterogeneity and periodicity (high in winter and low in summer). The PM concentration in Heilongjiang Province grows dramatically during the heating season (from October to April next year). Coal-fired heating not only consumes energy but also harms the environment. However, due to the Heilongjiang Province’s policies (Air Pollution Prevention and Control Action Plan) [10], the downward trend of annual PM concentration is obvious, especially after 2017, which indicates the progress in the prevention and control of air pollution in Heilongjiang. In the future, Heilongjiang Province should implement a multi-energy integrated heating system that makes full use of renewable energy sources such as wind, water, and solar energy. Additionally, it is critical to consider advancing science and technology, discovering methods to store renewable energy, and even storing energy in the summer for use in winter heating. Finally, increasing the rate of urban greening, focusing on forest ecosystem protection, and enhancing the ecosystem’s ecological benefits are all necessary.

## 5. Conclusions

In this study, the spatial-temporal heterogeneity of PM (PM_2.5_ and PM_10_) concentration in Heilongjiang Province during 2014–2018 and the key impacting factors were investigated based on PCA-based local and global models. Six PCs with a contribution rate of 87% were selected, and each PC’s (PC1–PC6) meaning was obvious after rotation, representing the temperature, wind speed, air pressure, atmospheric pollution, humidity, and vegetation cover, respectively. According to the model assessment, all the PCA-based local models (PCA-GWR, PCA-TWR, and PCA-GTWR) were superior to the PCA-based global (PCA-OLS) model, in which PCA-GTWR performed the best. Air pollutants (mainly SO_2_, NO_2_, and CO) have the largest influence on both PM_2.5_ and PM_10_. The temperature has the second greatest impact on PM_2.5_, followed by air pressure, wind speed, vegetation cover, and humidity factor, whereas the impacts of temperature, pressure, and vegetation cover on PM_10_ are roughly equivalent, followed by wind speed. In general, PM is positively related to other air pollutants (mainly SO_2_, NO_2_, and CO) and air pressure, and negatively correlated to temperature, wind speed, and vegetation cover.

This study demonstrated that the temporal heterogeneity was more pronounced than the spatial heterogeneity of PM in Heilongjiang Province. This work provides temporal and spatial heterogeneity evidence of the relationship between PM and meteorological factors, air pollutants, and vegetation cover. Furthermore, this work addresses both excessive impact factors and temporal and spatial heterogeneity and provides a theoretical foundation for precise local government prevention and control. 

## Figures and Tables

**Figure 1 ijerph-19-11627-f001:**
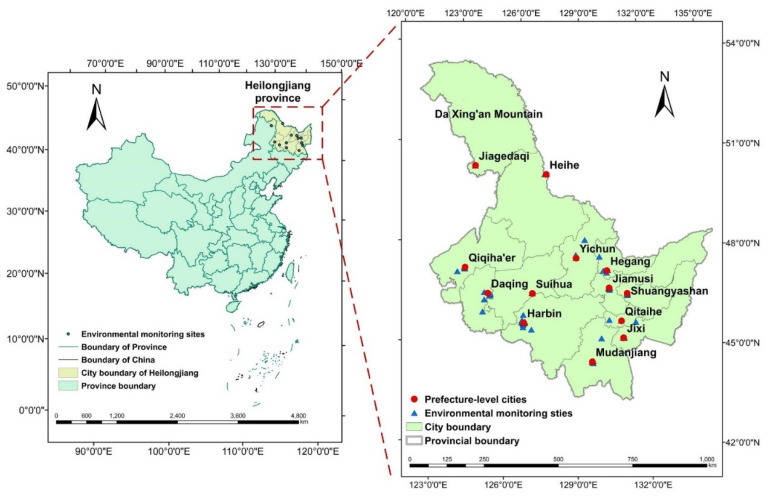
The location of Heilongjiang Province, People’s Republic of China (containing the Da Xing’an Mountain, Drawing review No: GS (2020)4619). Note: Jiagedaqi is a special residential neighborhood in which the Da Xing’an Mountain region’s environmental monitoring facility is located.

**Figure 2 ijerph-19-11627-f002:**
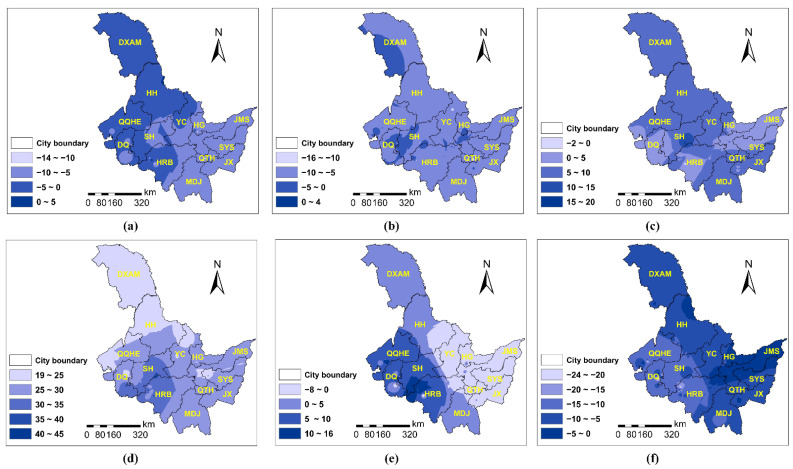
The PCA-GTWR parameter estimates of PM_2.5_ interpolated by Inverse Distance Weighted (IDW) method: (**a**) PC1 (temperature); (**b**) PC2 (wind speed); (**c**) PC3 (pressure); (**d**) PC4 (atmospheric pollutant); (**e**) PC5 (humidity); (**f**) PC6 (vegetation coverage). Note: HRB—Harbin; MDJ—Mudanjiang; QTH—Qitaihe; JX—Jixi; SYS—Shuangyashan; JMS—Jiamusi; HG—Hegang; YC—Yichun; SH—Suihua; DQ—Daqing; QQHE—Qiqiha’er; HH—Heihe; DXAM—Da Xing’an Mountain.

**Figure 3 ijerph-19-11627-f003:**
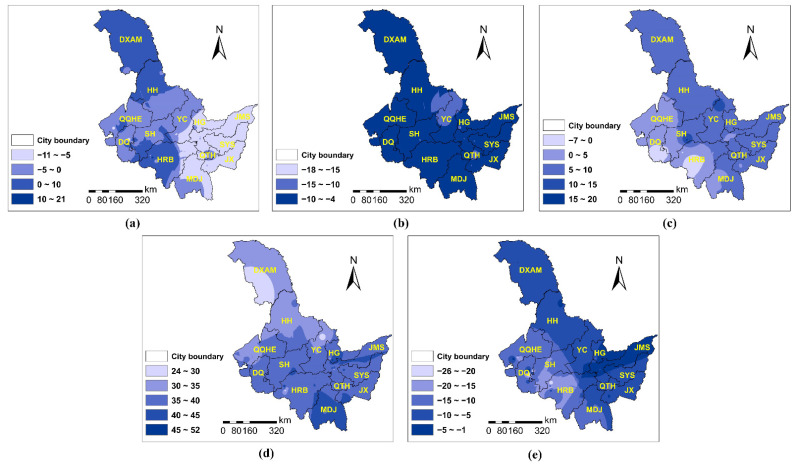
The PCA-GTWR parameter estimates of PM_10_ interpolated by IDW: (**a**) PC1 (temperature); (**b**) PC2 (wind speed); (**c**) PC3 (pressure); (**d**) PC4 (atmospheric pollutant); (**e**) PC6 (vegetation coverage). Note: HRB—Harbin; MDJ—Mudanjiang; QTH—Qitaihe; JX—Jixi; SYS—Shuangyashan; JMS—Jiamusi; HG—Hegang; YC—Yichun; SH—Suihua; DQ—Daqing; QQHE—Qiqiha’er; HH—Heihe; DXAM—Da Xing’an Mountain.

**Figure 4 ijerph-19-11627-f004:**
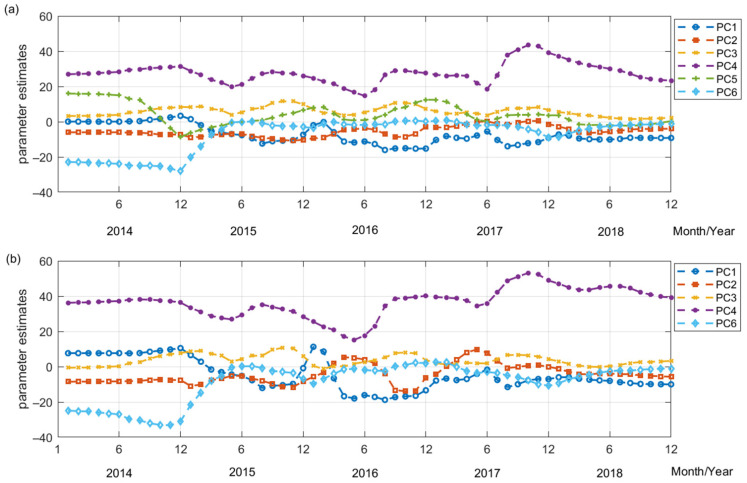
The temporal changes of PCA-GTWR parameter estimates for PM from 2014 to 2018: (**a**) PM_2.5_; (**b**) PM_10_. Note: PC1—temperature; PC2—wind speed; PC3—pressure; PC4—atmospheric pollutant; PC5—humidity; PC6—vegetation coverage.

**Figure 5 ijerph-19-11627-f005:**
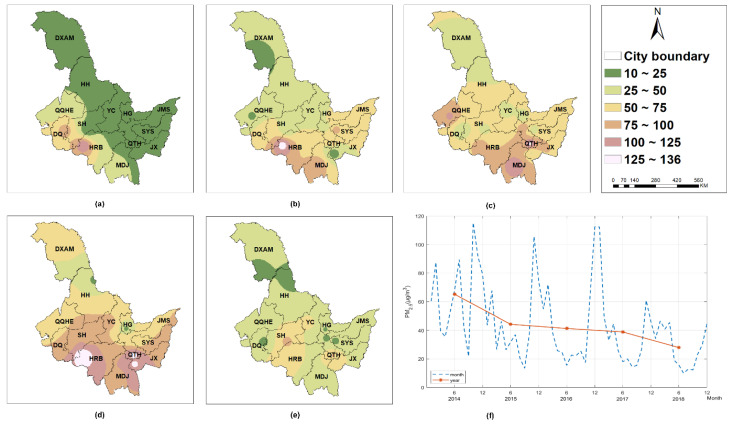
The spatial and temporal distribution of PM_2.5_ (μg/m^3^) in Heilongjiang Province from 2014 to 2018: (**a**) 2014; (**b**) 2015; (**c**) 2016; (**d**) 2017; (**e**) 2018; (**f**) temporal trend of annual concentration. Note: HRB—Harbin; MDJ—Mudanjiang; QTH—Qitaihe; JX—Jixi; SYS—Shuangyashan; JMS—Jiamusi; HG—Hegang; YC—Yichun; SH—Suihua; DQ—Daqing; QQHE—Qiqiha’er; HH—Heihe; DXAM—Da Xing’an Mountain.

**Figure 6 ijerph-19-11627-f006:**
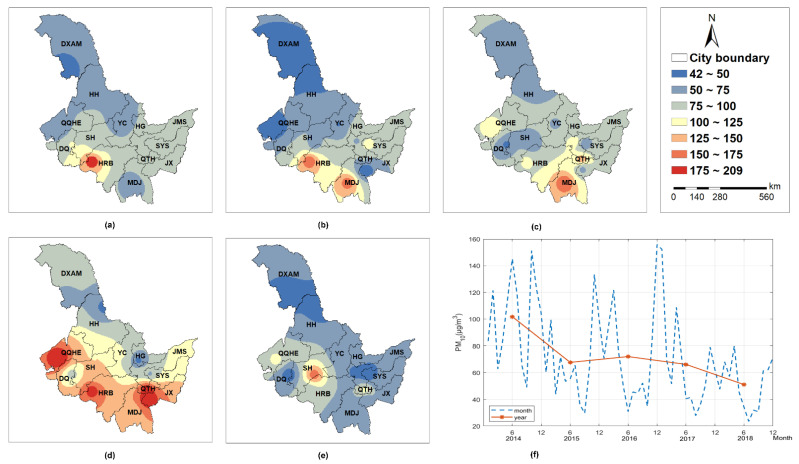
The spatial and temporal distribution of PM_10_ (μg/m^3^) in Heilongjiang Province from 2014 to 2018: (**a**) 2014; (**b**) 2015; (**c**) 2016; (**d**) 2017; (**e**) 2018; (**f**) temporal trend of annual concentration. Note: HRB—Harbin; MDJ—Mudanjiang; QTH—Qitaihe; JX—Jixi; SYS—Shuangyashan; JMS—Jiamusi; HG—Hegang; YC—Yichun; SH—Suihua; DQ—Daqing; QQHE—Qiqiha’er; HH—Heihe; DXAM—Da Xing’an Mountain.

**Table 1 ijerph-19-11627-t001:** Descriptive statistics of the variables used in this study.

Category	Variables	Min	Mean	Max	SD	Kurtosis	Skewness
Dependent variable	PM_2.5_ (μg/m^3^)	4.00	41.08	262.00	37.17	6.59	2.35
	PM_10_ (μg/m^3^)	10.00	66.74	341.00	46.82	3.95	1.85
Air pollutants	SO_2_ (μg/m^3^)	2.00	20.66	191.00	21.34	13.80	3.03
	NO_2_ (μg/m^3^)	2.00	28.37	101.00	16.15	1.94	1.31
	CO (mg/m^3^)	0.00	0.78	5.00	0.45	9.46	2.24
	O_3_ (μg/m^3^)	4.00	73.71	215.00	29.01	1.13	0.92
Meteorological factors	ARH	0.24	0.62	0.94	0.13	−0.13	−0.37
	HCP (mm)	0.00	0.50	23.64	1.76	41.32	5.78
	SH(h)	0.00	7.81	13.93	2.99	−0.16	−0.40
	MaxP (hPa)	951.36	996.44	1029.09	11.83	−0.04	−0.28
	AP (hPa)	947.56	993.94	1026.41	11.98	0.00	−0.28
	MinP (hPa)	943.96	991.08	1021.70	12.13	0.02	−0.31
	MaxT (°C)	−26.57	7.47	36.06	15.81	−1.38	0.00
	AT (°C)	−36.47	1.13	28.70	15.57	−1.33	−0.02
	MinT (°C)	−43.67	−4.86	23.72	15.31	−1.25	−0.01
	MaxWS (m/s)	1.77	5.58	13.03	1.73	0.52	0.60
	AWS (m/s)	0.56	2.64	7.23	1.03	1.01	0.85
	EWS (m/s)	2.59	8.97	20.60	2.88	0.12	0.49
	MaxST (°C)	−16.33	17.34	63.06	21.19	−1.39	0.24
	AST (°C)	−18.09	6.01	36.37	14.39	−1.25	0.35
	MinST (°C)	−23.40	−1.22	23.36	10.43	−0.92	0.42
Vegetation coverage	NDVI	−0.31	−0.07	0.78	0.12	9.08	2.73

Note: ARH—daily average relative humidity; HCP—daily cumulative precipitation; SH—daily sun hours; MaxP—daily maximum air pressure; AP—daily average air pressure; MinP—daily minimum air pressure; MaxT—daily maximum temperature; AT—daily average temperature; MinT—daily minimum temperature; MaxWS—daily maximum wind speed; AWS—daily average wind speed; EWS—daily extreme wind speed; MaxST—daily maximum surface temperature; AST—daily average surface temperature; MinST—daily minimum surface temperature.

**Table 2 ijerph-19-11627-t002:** Principal component loadings of each PC after the maximum variance orthogonal rotation approach.

PCs	PC1	PC2	PC3	PC4	PC5	PC6
SO_2_ (μg/m^3^)	−0.36	−0.10	0.09	** *0.77* **	0.03	0.04
NO_2_ (μg/m^3^)	−0.07	−0.20	0.24	** *0.85* **	−0.02	−0.15
CO (mg/m^3^)	−0.23	−0.19	0.15	** *0.72* **	0.09	−0.07
O_3_ (μg/m^3^)	0.56	0.10	−0.14	−0.11	−0.46	−0.27
ARH	0.03	−0.46	0.02	0.14	** *0.74* **	0.08
HCP (mm)	0.25	0.05	−0.13	−0.12	** *0.69* **	−0.08
SH (h)	0.49	−0.06	0.00	−0.18	**−0.69**	0.09
MaxP (hPa)	−0.27	−0.02	** *0.94* **	0.18	−0.04	−0.06
AP (hPa)	−0.26	−0.08	** *0.94* **	0.17	−0.04	−0.05
MinP (hPa)	−0.25	−0.13	** *0.94* **	0.16	−0.03	−0.05
MaxT (°C)	** *0.95* **	0.05	−0.21	−0.14	−0.06	0.07
AT (°C)	** *0.95* **	0.07	−0.18	−0.17	0.00	0.07
MinT (°C)	** *0.94* **	0.10	−0.16	−0.18	0.09	0.07
MaxWS (m/s)	0.02	** *0.97* **	−0.11	−0.12	−0.04	−0.01
AWS (m/s)	0.00	** *0.94* **	0.02	−0.15	−0.06	0.00
EWS (m/s)	0.12	** *0.95* **	−0.13	−0.16	−0.06	0.01
MaxST (°C)	** *0.94* **	0.01	−0.19	−0.17	−0.12	0.06
AST (°C)	** *0.96* **	0.00	−0.19	−0.14	−0.02	0.07
MinST (°C)	** *0.92* **	−0.02	−0.18	−0.12	0.18	0.08
NDVI	0.20	−0.02	−0.12	−0.14	−0.04	** *0.92* **
Meaning of PCs (Cumulative Proportion)	**Temperature** (40%)	**Wind speed** (57%)	**Pressure** (67%)	**Atmospheric pollutant****s** (75%)	**Humidity** (82%)	**Vegetation cover** (87%)

**Table 3 ijerph-19-11627-t003:** PCA-OLS model’s parameter estimations for PM (*p*< 0.01).

PM2.5	Estimate	*t*-Test	*p*-Value	PM10	Estimate	*t*-Test	*p*-Value
Intercept	41.08	136.27	<2 × 10^−16^	Intercept	66.74	162.13	<2 × 10^−16^
PC1	−7.39	−24.50	<2 × 10^−16^	PC1	−4.60	−11.18	<2 × 10^−16^
PC2	−5.18	−17.17	<2 × 10^−16^	PC2	−1.83	−4.45	8.81× 10^−6^
PC3	6.85	22.72	<2 × 10^−16^	PC3	4.72	11.47	<2 × 10^−16^
PC4	26.41	87.60	<2 × 10^−16^	PC4	33.43	81.20	<2 × 10^−16^
PC5	2.68	8.88	<2 × 10^−16^	PC6	−4.67	−11.33	<2 × 10^−16^
PC6	−3.03	−10.05	<2 × 10^−16^				
	$R_{a}^{2}$	0.61			$R_{a}^{2}$	0.54	
	AICc	54,275.10			AICc	57,976.60	

**Table 4 ijerph-19-11627-t004:** PCA-GWR, PCA-TWR, and PCA-GTWR parameter estimates of PM_2.5_ (the models are fitted by an adaptive neighbor selection algorithm).

Models	Variables	Min	Mean	Max	SD	Kurtosis	Skewness	Model Fitting Information
PCA-GWR (Num of Neighbors = 1041)	intercept	7.43	35.76	48.95	15.09	−0.64	−1.04	$R_{a}^{2}$: 0.68 AICc: 53,295.1
	PC1	−13.61	−5.75	−0.18	4.29	−1.37	−0.25	
	PC2	−6.46	−3.88	−0.98	1.81	−1.42	0.14	
	PC3	2.05	12.01	26.14	8.40	−1.02	0.70	
	PC4	15.17	30.05	37.55	5.74	0.68	−0.95	
	PC5	−2.74	3.63	8.86	2.88	−0.83	0.11	
	PC6	−21.18	−7.64	−1.48	6.06	−1.13	−0.71	
PCA-TWR (Num of Neighbors = 595)	intercept	5.18	39.68	95.87	13.87	4.50	0.87	$R_{a}^{2}$: 0.75 AICc: 51,804.3
	PC1	−25.34	−3.72	53.83	12.88	6.66	1.98	
	PC2	−14.13	−4.17	8.21	4.40	−0.01	0.09	
	PC3	−12.00	5.08	12.97	4.60	2.39	−0.98	
	PC4	1.20	22.77	59.16	11.46	1.07	0.67	
	PC5	−9.79	5.35	24.28	7.48	−0.46	0.67	
	PC6	−32.19	−4.41	6.96	7.63	3.01	−1.83	
PCA-GTWR (Num of Neighbors = 595)	intercept	23.64	41.86	59.43	6.47	−0.56	−0.04	$R_{a}^{2}$: 0.76 AICc: 51,607.2
	PC1	−27.94	−8.35	22.70	5.56	0.49	0.37	
	PC2	−19.60	−4.64	10.77	4.41	1.71	0.13	
	PC3	−6.68	5.90	26.56	4.83	2.50	0.85	
	PC4	8.64	28.79	51.43	8.41	−0.24	0.43	
	PC5	−13.17	3.37	22.37	5.90	1.15	0.88	
	PC6	−35.30	−5.04	8.47	7.56	2.89	−1.87	

**Table 5 ijerph-19-11627-t005:** PCA-GWR, PCA-TWR, and PCA-GTWR parameter estimates of PM_10_ (the models are fitted by an adaptive neighbor selection algorithm).

Models	Variables	Min	Mean	Max	SD	Kurtosis	Skewness	Model Fitting Information
PCA-GWR (Num of Neighbors = 937)	intercept	27.39	62.51	85.65	19.74	−0.88	0.26	$R_{a}^{2}$: 0.63 AICc: 56,894.4
	PC1	−12.54	−2.01	9.20	6.36	−0.89	0.08	
	PC2	−8.22	−0.14	6.69	4.75	−1.34	0.06	
	PC3	−1.77	11.45	25.94	8.29	−0.65	0.11	
	PC4	23.41	40.44	45.95	6.13	2.02	0.08	
	PC6	−20.11	−8.67	−2.41	4.64	−0.83	0.06	
PCA-TWR (Num of Neighbors = 602)	intercept	0.56	66.42	155.0	22.90	6.32	1.38	$R_{a}^{2}$: 0.71 AICc: 55,447.7
	PC1	−24.92	1.12	84.12	20.42	6.80	2.36	
	PC2	−15.84	−3.22	21.88	8.36	0.46	0.87	
	PC3	−21.35	3.70	14.46	6.36	4.30	−1.70	
	PC4	1.36	30.63	69.10	12.93	0.70	0.38	
	PC6	−32.82	−5.32	11.66	8.26	2.70	−1.54	
PCA-GTWR (Num of Neighbors = 595)	intercept	39.12	69.63	114.6	11.16	−0.63	0.18	$R_{a}^{2}$: 0.73 AICc: 55,114.7
	PC1	−31.98	−5.83	45.62	8.41	2.03	0.79	
	PC2	−21.94	−2.82	22.36	6.87	0.83	0.52	
	PC3	−13.21	3.65	27.61	6.36	1.16	0.18	
	PC4	9.26	38.04	60.40	10.78	−0.43	−0.10	
	PC6	−36.97	−6.44	12.97	8.81	2.37	−1.60	

**Table 6 ijerph-19-11627-t006:** Comparison of PCA-OLS, PCA-GWR, PCA-TWR, PCA-GTWR.

Model	PM_2.5_				PM_10_			
	AICc	$R_{a}^{2}$	RMSE (μg/m^3^)	MAE (μg/m^3^)	AICc	$R_{a}^{2}$	RMSE (μg/m^3^)	MAE (μg/m^3^)
PCA-OLS	54,275.10	0.61	23.23	0.30	57,976.60	0.54	31.72	0.41
PCA-GWR	53,295.10	0.68	21.16	0.27	56,892.60	0.63	28.62	0.37
PCA-TWR	51,804.30	0.75	18.59	0.24	55,836.90	0.69	26.16	0.34
PCA-GTWR	51,607.20	0.76	18.20	0.24	55,365.10	0.71	25.04	0.33

## Data Availability

The data that support the findings of this study are available from the corresponding author, (Z.Z. or Q.W.), upon reasonable request.

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
