# Peer review of "Spatiotemporal Heterogeneity and the Key Influencing Factors of PM2.5 and PM10 in Heilongjiang, China from 2014 to 2018"

_ijerph, 2022, doi:10.3390/ijerph191811627_

Round 1

Reviewer 1 Report

The study investigates the spatial–temporal heterogeneity of particulate matter concentration in Heilongjiang Province in China using Principal Component Analysis (PCA) and other PCA-based models, at a local and global scales. The research is interesting and is within the scope of International Journal of Environmental Research and Public Health.

PCA was considered as a based model, used to fit other models as OLS based, geographical, temporal, and geographical-temporal models. PCA seems to required according to the authors a correlation of independent variables. However, correlation of independent variables is to avoid for the other models. Unfortunately, it is not clear how PCA is extended to other models, owing to those conflicting conditions. Also, the method section does not clearly explain the grouping of all listed models in terms of local and global models. I also found some typos here and there.

These are general comments, and you will find in the following, more specific comments to be addressed.

L39 Do you mean human or plant mortality? Important to specify because in the earlier sentences (L34-37), you suggested that PM influences both plant photosynthesis and human health.

L56 “there has been” refers to “studies”, then the plural form is “there have been”.

L103. Please replace “explain” with “explains”, if related to “study” in L101.

L136-139 We would expect a brief description of the Environmental measurement procedure and design, or at least a reference to a study describing them.

L147 “maximum wind speed” listed twice. Do you rather mean “minimum”?

L173 All meteorological variables have maximum, average and minimum values. For “wind speed” maximum and average data are also provided. However instead of “minimum” you rather stated “extreme” wind speed. Please explain, what you mean by “extreme wind speed. Also, in the table 2 itself, the header indicates “maximum” wind speed.

L 175 The method section is confusing and not easy to follow. All models are listed but what is missing is their grouping in terms of local and global models. This section requires a thorough edit to increase its clarity. I would suggest a paragraph after L175 explaining the grouping in local and global models and the rational behind such a grouping.

L183 What the subscript “m” in “xm” stands for?

L202-203. The Ep. 6 seems to have a problem. I don’t see the subscript “i”. Please clarify.

L206 I would say “error term” instead of “residual”.

L288 Please replace “exclude” with “excluded”.

L398 multicollinearity occurs when independent variables are correlated between them. In L278-279, you stated that variables are highly correlated, as required by PCA. These two conflicting statements make me wondering why you suggest here that PCA removes the multicollinearity. First goal of PCA is to select only most significant variables based on their loading. Obviously, autocorrelation is less likely to occur when the number of variables decreases, but the remaining variable may be correlated.

L423 May influence positively or negatively?

L 424 Tables 2 &3 show a negative relationship between temperature and PM, which means that the higher the temperature, the lower the PM. Based on that, the statement according which the vegetation decreases the temperature resulting in PM decrease is hardly convincing.

L428-430 PCA-OLS model derived from PCA model. If in one hand autocorrelation of independent variables are required for the PCA (see L278-279), and in the other hand If the autocorrelation may violate the assumption of PCA-OLS model, I’m wondering how you deal with these two conflicting conditions.

Author Response

Thank you for your careful review and comments. Please see the attachment.

Reviewer 2 Report

Please see the attached pdf file.

Author Response

(The authors gave the same response as above.)
